# Regulation of molecular transport in polymer membranes with voltage-controlled pore size at the angstrom scale

Yuzhang Zhu [1], Liangliang Gui[1], Ruoyu Wang [2], Yunfeng Wang[1], Wangxi Fang [1], Menachem Elimelech [3], Shihong Lin [2,4] ✉ & Jian Jin [1,5] ✉

Polymer membranes have been used extensively for Angstrom-scale separation of solutes and molecules. However, the pore size of most polymer membranes has been considered an intrinsic membrane property that cannot be adjusted in operation by applied stimuli. In this work, we show that the pore size of an electrically conductive polyamide membrane can be modulated by an applied voltage in the presence of electrolyte via a mechanism called electrically induced osmotic swelling. Under applied voltage, the highly charged polyamide layer concentrates counter ions in the polymer network via Donnan equilibrium and creates a sizeable osmotic pressure to enlarge the free volume and the effective pore size. The relation between membrane potential and pore size can be quantitatively described using the extended Flory-Rehner theory with Donnan equilibrium. The ability to regulate pore size via applied voltage enables operando modulation of precise molecular separation in-situ. This study demonstrates the amazing capability of electro-regulation of membrane pore size at the Angstrom scale and unveils an important but previously overlooked mechanism of membrane-water-solute interactions.

The ability to regulate membrane pore size as needed has been a topic of active research for developing next-generation stimuli-responsive smart membranes[1,2]. Current pore regulation strategies primarily rely on temperature, light, and solution composition to induce configurational changes of polymers grafted into the membrane pores or at the membrane surface[1,3]. Pore size regulation is typically achieved via contraction or dilation of the grafted polymers upon their exposure to stimuli. The size of pores that can be regulated by stimuli-induced configurational changes of grafted polymers ranges from nanometers[4] to micrometers[5]. For solute separation that requires Angstrom-level pores (e.g., nanofiltration and reverse osmosis), stimuli-driven operando pore size regulation is rare with a

very recent example achieved using photo-isomerization of a covalent organic framework[6].

Polyamide (PA) membranes are the state-of-the-art solute separation membranes that dominate the market of desalination and water reuse[7,8]. Forming from a reaction between amine and acid chloride monomers at the water–oil interface, the PA layer is a cross-linked semi-aromatic and/or aromatic network with a thickness <100 nm[9]. The effective pore size can be tuned in membrane synthesis via adjusting multiple factors such as monomer species[10] and concentrations[11], additives[12,13], and fabrication conditions[14,15], but has been considered as fixed and an intrinsic membrane property once the PA layer is formed.

[1]i-Lab, Suzhou Institute of Nano-Tech and Nano-Bionics, Chinese Academy of Sciences, Suzhou 215123, PR China. [2]Department of Civil and Environmental Engineering, Vanderbilt University, Nashville, TN 37235, USA. [3]Department of Chemical and Environmental Engineering, Yale University, New Haven, CT 06520-8286, USA. [4]Department of Chemical and Biomolecular Engineering, Vanderbilt University, Nashville, TN 37235, USA. [5]College of Chemistry, Chemical Engineering and Materials Science, Innovation Center for Chemical Science & Jiangsu Key Laboratory of Advanced Functional Polymer Design and Application, Soochow University, Suzhou 215123, PR China. ✉e-mail: shihong.lin@vanderbilt.edu; jjin@suda.edu.cn

The PA layer is a charged polymer that can undergo swelling and deswelling depending on the solution and membrane conditions. A recent study by Hedge et al. has proposed a "fluid-solid" two-phase model[16] that considers the PA layer as a hydrogel that de-swells in an electrolyte solution (Fig. 1a). Under such a model, the de-swollen PA layer contracts to reduce the free volume (under the solution-diffusion model) or effective pore size (under the pore flow model). The PA layer de-swells in reverse osmosis because most conventional PA membranes are only weakly charged and recent study suggests that only the surface of the PA layer is charged while most of the PA matrix is uncharged[17,18].

If the membrane is strongly charged, however, Donnan equilibrium can lead to a much higher concentration of counter ions in the polymer network (Fig. 1b). The development of concentration difference within and outside the polymer network also creates an osmotic pressure difference that swells the membrane. As this osmotic swelling is caused by osmosis into a polymer network with a high electrical potential, we herein refer to this phenomenon as electrically induced osmotic swelling. Electrically induced osmotic swelling can be leveraged for modulating the effective pore size if the electrical potential of the membrane can be adjusted in operation (Fig. 1c). Therefore, operando regulation of molecular separation may be achieved with an electrically conductive membrane with Angstrom-scale pores.

Herein, we report the observation of operando regulation of the pore size of an electrically conductive PA membrane based on the proposed mechanism of electrically induced osmotic swelling. We first fabricate an electrically conductive PA membrane using carbon nanotubes as the conductive support layer. We then perform a series of experiments using this conductive PA membrane to evaluate its effective pore size under different applied potentials and solution conditions. We also develop a quantitative model using the extended Flory–Rehner theory to relate the effective pore size to the difference between intrapore and bulk osmotic pressures. Finally, we discuss the implication of electrically induced osmotic swelling beyond pore size regulation and how our findings extend the current theoretical framework of water and solute transport in nanofiltration.

## Results

### Electrically conductive polyamide membrane

Thin-film composite polyamide (TFC-PA) nanofiltration (NF) membrane was fabricated by interfacial polymerization with piperazine (PIP) and 1,3,5-trimesoyl chloride (TMC) on the rough surface of a network of hydroxyl-functionalized multiwalled carbon nanotubes (referred to as CNTs) (Supplementary Fig. 1). The aqueous PIP solution was trapped in the CNTs network via capillary, forming a concave water/hexane interface within the network upon introducing the hexane solution of TMC to the surface of this CNTs network impregnated with PIP solution. The reaction between PIP and TMC results in a PA layer embedded in CNTs network (Fig. 2a).

Comparing the scanning electron microscopy (SEM) images of the CNTs network surface before and after interfacial polymerization (Supplementary Fig. 1 vs. Fig. 2b) reveals complete filling of the interstitial voids near the surface of the CNTs network. Transmission electron microscopy (TEM) images of the cross-section of the CNTs-PA network confirms the formation of an ultrathin PA layer near the surface of the ~500 nm-thick CNTs network (Fig. 2c), which is consistent with the observation from the cross-sectional SEM image of the CNTs-PA network (Supplementary Fig. 2). High-resolution cross-sectional TEM image verifies the penetration of the interconnected CNTs network into the PA layer (Fig. 2d), which renders the PA layer electrically conductive. The electrical resistance of the CNTs-PA membrane is slightly higher than that of a pristine CNTs network but orders of magnitude lower than that of a CNTs-free PA membrane (Fig. 2e).

### Precise control of Angstrom-level pore size by applied voltage

The high electrical conductivity enables the CNTs-PA membrane to be used as a working electrode in an NF process with the counter electrode immersed in the feed solution (Fig. 3a). SEM images (Supplementary Fig. 3) and surface compositional analysis using X-ray photoelectron spectroscopy (Supplementary Fig. 4) reveal little compositional change of the CNT-PA surface when it was used as a cathode up to an applied voltage of 1.0 V. However, the CNT-PA membrane was subject to irreversible chemical degradation when 1.5 V was applied (Supplementary Fig. 5). The molecular weight cutoff (MWCO) (i.e., the interpolated molecular weight of neutral poly (ethylene glycol) (PEG) molecules with a 90% rejection) of the CNTs-PA membrane was evaluated using a standard method of measuring the rejections of PEG with different molecular weights (Fig. 3b). From the PEG rejections, the pore radius ($r_p$) of CNTs-PA was determined by fitting the PEG rejection data with the hindered transport model[19–21], which applies to NF where diffusion dominates advection for solute transport.

In the presence of electrolyte (e.g., $Na_2SO_4$), the pore size distribution systematically shifted with increasing the applied voltage (Fig. 3b). Specifically, the MWCO increased from 361 Da without applied voltage to 475 Da with 1.0 V applied. The ability to systematically regulate pore size using applied voltage was further confirmed with a voltage scan from 0 to 1.0 V with a 0.1 V increment (Fig. 3c and Supplementary Fig. 6), which shows the amazing capability of

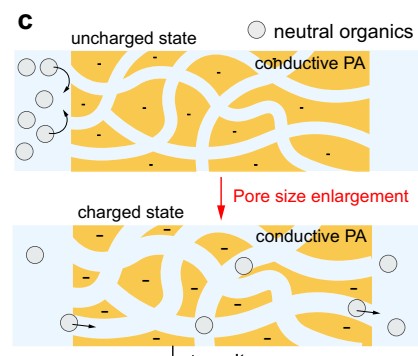

**Fig. 1 | Theory of electrically induced osmotic pore size regulation. a** De-swelling of a weakly charged PA network exposed to an electrolyte solution. Osmosis drives water out of the polymer network "into" the electrolyte solution, causing polymer de-swelling. **b** Electrically induced osmotic swelling of a strongly charged PA network exposed to an electrolyte solution. Donnan equilibrium results in a heightened concentration of ions compared to that of the bulk, resulting in electrostatically-induced osmotic swelling. **c** A hypothesized strategy of pore size regulation by applying a voltage to an electrically conductive polymer network in the presence of electrolyte (electrolyte not shown in the schematics).

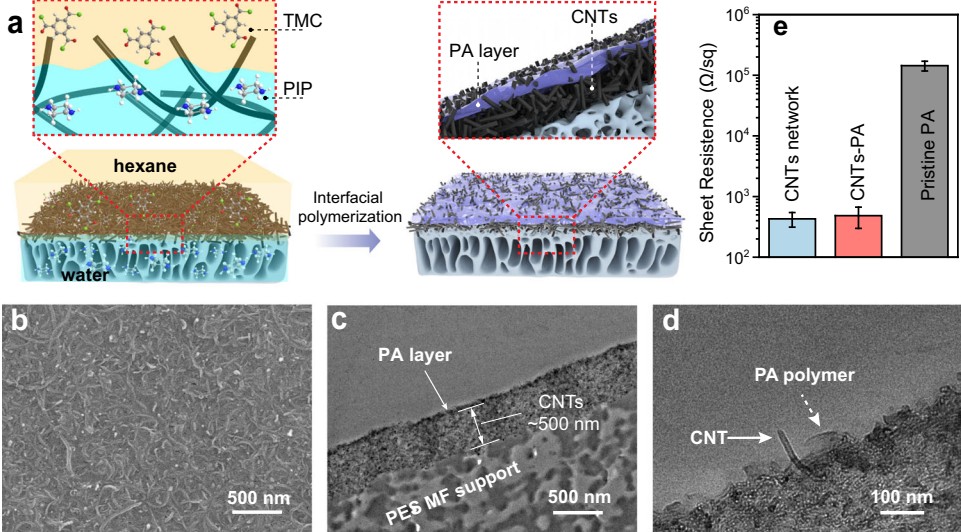

**Fig. 2 | Fabrication, structure, and electrical resistance of CNTs-PA membrane.** **a** Schematic illustration of the approach for fabricating electrically conductive CNTs-PA membrane. **b** Top-view SEM image of the CNTs-PA membrane. **c**, **d** Cross-sectional TEM images of the CNTs-PA membrane. **e** Sheet resistances of the PA-free CNTs network, the CNTs-PA membrane, and the pristine (CNTs-free) PA membrane. Error bar represents the standard deviation of three replicate measurements. Source data are provided as a Source Data file.

operando regulation of Angstrom-scale pore size with a sub-Angstrom precision.

The change of pore size at a certain applied voltage also increased with increasing Na$_2$SO$_4$ concentration (Fig. 3d and Supplementary Fig. 7). Similar pore size enlargement at an increased applied voltage was observed with different electrolytes (Supplementary Fig. 8 and Supplementary Table 1). Importantly, the operando tuning of pore size by applied voltage is reversible (Fig. 3e and Supplementary Fig. 9). Besides imparting membrane reusability in practical applications, the reversibility in pore tuning also suggests that the pore size is a state function of applied voltage, i.e., in the voltage window that does not induce any irreversible chemical reaction to the CNTs-PA layer, the pore size of the CNTs-PA is a function of applied voltage with a given feed water composition. Notably, similar phenomena of pore size regulation by applied voltage can also be achieved using conductive PA membranes forming on a network of silver nanowires instead of CNTs (Supplementary Figs. 10–12). All these observations suggest that the mechanism of pore size regulation is purely physical, supporting the hypothesis of electrically induced osmotic swelling as the core mechanism for pore size regulation.

The ability to regulate pore size via applied voltage enables operando modulation of molecular separation in situ. For instance, we applied the CNTs-PA membrane to filter a feed solution comprising two dyes with different molecular weights: acid fuchsin (AF, MW = 585.5 Da) and methyl orange (MO, MW = 327.3 Da). Without an applied voltage, the rejections of AF and MO were 99.4% and 95.2%, respectively (Supplementary Fig. 13). Applying a voltage of 1.0 V reduced the rejection of MO to 20.5%, whereas inducing almost no change to AF rejection. Leveraging the voltage-dependent solute rejections, we can apply the same NF membrane to attain either separation of solvent from both solutes or different solutes from each other (Fig. 3f). In addition, the ability to regulate pore size via applied voltage enables the on-off adjustment of salt rejection and permeation for desalination. As shown in Supplementary Fig. 14a, the Na$_2$SO$_4$ rejection decreased from 98.5% to 90% when the applied voltage changed from 0 to 1 V using 1000 ppm Na$_2$SO$_4$ as feed. The corresponding permeating flux increased from 42 to 65 Lm$^{-2}$h$^{-1}$ simultaneously. The substantial enhancement in the permeating flux is mainly attributed to the enlargement of the pore size. In contrast, the slight change in the Na$_2$SO$_4$ rejection is attributed to the enhancement of negative surface

potential by the voltage, which partially offsets the adverse effect of pore enlargement. The rejection and permeating flux can be easily on-off tuned by the voltage, as evidenced by switching the voltage between 0 and 1 V 10 times in the cycling experiment (Supplementary Fig. 14b).

## Discussion
### Mechanism and model of electrically induced osmotic pore size regulation

When an external potential is applied to the conductive CNTs-PA layer, counter ion partitioning into the PA layer becomes favorable via the Donnan effect. Consequently, the ion concentration and thus osmotic pressure become higher in the CNTs-PA matrix than in the feed solution (Fig. 1b). According to the extended Flory-Rehner theory with Donnan equilibrium for polymer network swelling, the total osmotic pressure of the PA layer arises from polymer mixing, polymer elasticity, and the Donnan contribution, $\Delta\pi_D$[22,23]. The Donnan contribution to the swelling pressure stems from the potential-induced difference in ion concentrations inside and outside the polymer network. For a conductive CNTs-PA membrane, the Donnan contribution can be modulated by an externally applied voltage to increase the osmotic pressure within the PA network.

Based on the Donnan equilibrium, the partition coefficient of counter ions to a matrix at a relatively high Donnan potential (e.g., > 100 mV) can far exceed unity. Let us consider a general electrolyte $M_{\nu^+}X_{\nu^-}$ that can fully dissociate to become $\nu^+$ cations $M^{z^+}$ carrying positive charges of $z^+|e|$ and $\nu^-$ anions $X^{z^-}$ carrying negative charges of $z^-|e|$. We apply the simplified Donnan model to estimate the contribution to the osmotic pressure difference between the bulk solution and the polymer matrix (i.e., the osmotic swelling pressure, $\Delta\pi_D$) induced by a Donnan potential[24,25], $\Delta\psi_D$:

$$\Delta\pi_D = \Delta cRT = RTc_i\left[\nu^+\exp\left(-\frac{z^+|e|\Delta\psi_D}{kT}\right) + \nu^-\exp\left(-\frac{z^-|e|\Delta\psi_D}{kT}\right) - \nu^+ - \nu^-\right]$$
(1)

where $R$ is the ideal gas constant, $k$ is the Boltzmann constant, $T$ is the absolute temperature, $\Delta c$ is the difference in ion concentration inside and outside the CNTs-PA network, and $c_i$ is the interfacial electrolyte

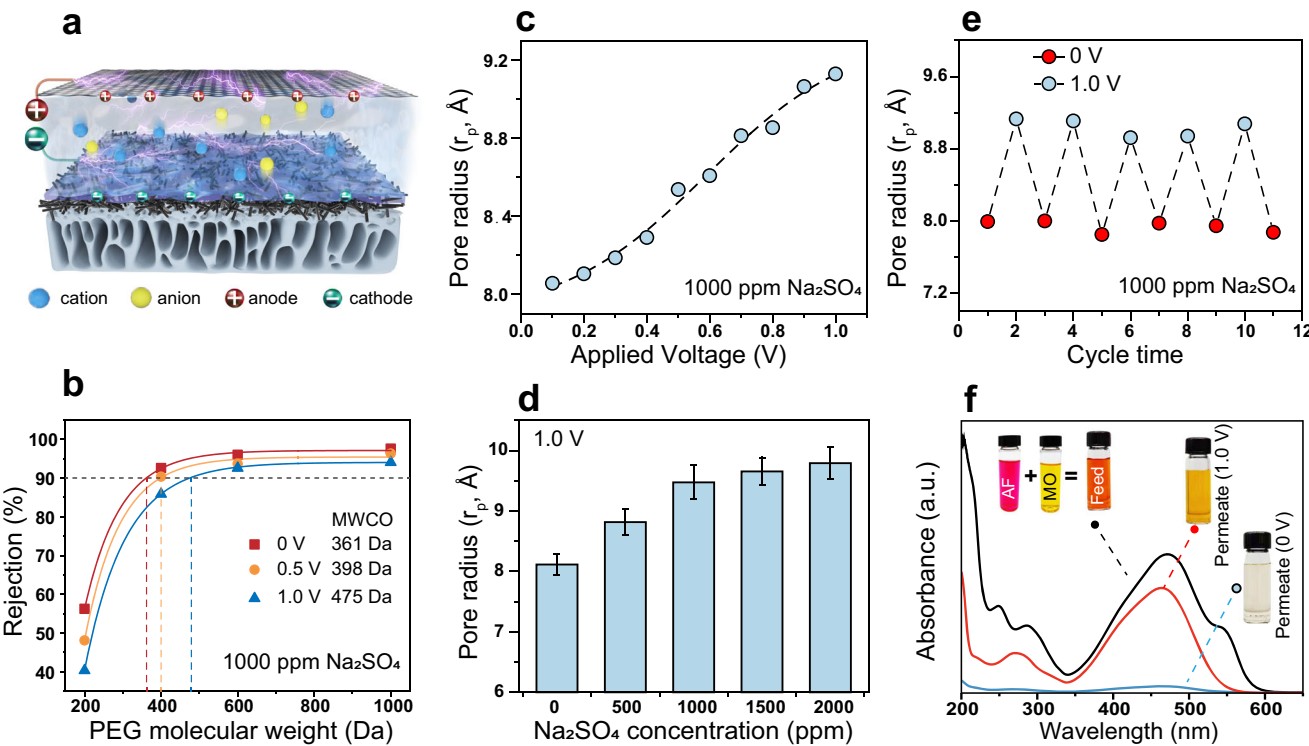

**Fig. 3 | Regulation of CNTs-PA membrane pore size via applied voltage.**
**a** Schematic illustration of nanofiltration using CNTs-PA membrane as the cathode.
**b** PEG rejections as function of molecular weight by the CNTs-PA membrane and corresponding MWCO at different applied voltages in the presence of 1000 ppm Na₂SO₄. **c** Pore radius of CNTs-PA membrane as a function of applied voltage in the presence of 1000 ppm Na₂SO₄. **d** Pore radius of CNTs-PA membrane as a function of Na₂SO₄ concentration at an applied voltage of 1.0 V. **e** Reversible change of pore radius of the CNTs-PA membrane when applied voltage switches between 0 and 1.0 V. **f** Demonstration of operando regulation of dye separation performance via adjusting the voltage applied on the CNTs-PA membrane. For **c**–**e**, the pore radii are estimated using the hinder transport model with rejections of PEG of different molecular weights as the inputs. Error bar represents the standard deviation of three replicate measurements. Source data are provided as a Source Data file.

concentration which may be higher than the bulk concentration when considering concentration polarization.

To evaluate $\Delta\psi_D$, we measured the electrical potential of the CNT-PA membrane ($\psi_m$) vs. a reference electrode (Ag/AgCl) when different voltages were applied between the working and counter electrodes. We focused on using the CNTs-PA membrane as a negative electrode because cyclic voltammetry revealed that the capacitance of the membrane is negligibly small when the CNTs-PA was used as a positive electrode (Supplementary Figs. 15–17). The measured electrical potential of the CNTs-PA membrane was stable and corresponded to the applied voltage (Supplementary Fig. 15a and Fig. 16), which allows us to estimate $\psi_m$. We estimated the Donnan potential ($\Delta\psi_D$) using the membrane potential ($\psi_m$) and the double layer capacitance $C_{DL}$ estimated from cyclic voltammogram (Supplementary Fig. 14), based on the theory that $\psi_m$ equals the sum of $\Delta\psi_D$ and the Stern potential $\Delta\psi_S$ when the system is at equilibrium[24,25].

The Donnan contribution ($\Delta\pi_D$) to the swelling pressure, the part of swelling pressure stemming from the $\Delta c$, correlates with the pore radius (Fig. 4a), lending strong support to the mechanism of the pore size regulation via electrically induced osmotic swelling. Such a mechanism is supported by additional evidence. For instance, when the CNTs-PA membrane was used as the positive electrode, little change of effective pore size was observed upon applied voltage (Supplementary Fig. 18) due to the limited capacitance (Supplementary Fig. 17). When a different amine monomer, methylpiperazine (MPIP), was used to fabricate the CNTs-PA membrane (Supplementary Figs. 19–22), the resulting membrane has little capacitance as compared to the PIP-based CNTs-PA membrane (Supplementary Fig. 21). Consequently, with an applied voltage of 1.0 V, much smaller changes in MWCO were observed with the CNTs-PA membrane based on

methyl piperazine as compared with that based on PIP (Supplementary Fig. 22). Moreover, when the feed solution contains no electrolyte, applied voltage had almost no impact on effective pore size (Supplementary Fig. 23), which suggests that the electrically driven swelling is not a Coulombic effect but relies on osmotic pressure and requires the presence of electrolyte. In fact, the strong dependence of the effective pore size on electrolyte concentration (Fig. 2d) also supports the mechanism of electrically induced osmotic swelling.

We apply the extended Flory–Rehner theory with Donnan equilibrium to relate the voltage-induced Donnan contribution to the osmotic pressure ($\Delta\pi_D$, as described by Eq. 1) and the measured change in pore size. While the effective pore radius measured using PEG rejections is a surface property, we assume the PA layer to be uniform and thus the inner pores, regardless of geometry, also have the same pore radius, $r_p$. We assume that the pore volume fraction (occupied by water), $\phi_w$, is proportional to the cube of pore radius (i.e., $\phi_w = \alpha r_p^3$, with $\alpha$ being a fitting parameter). The relation between $r_p$ and $\Delta c$ (estimated using Eq. 1) is described as follows:

$$\frac{\delta r_p}{\delta\Delta c} = \frac{\upsilon_w r_p}{\phi_w}\left[-\frac{1}{\phi_w} + 1 + 2\chi\phi_p + \frac{1}{N}\left(\frac{\phi_p^{-\frac{2}{3}}}{3} - \frac{1}{2}\right)\right]^{-1} \quad (2)$$

where $\upsilon_w$ is the molar volume of water, $\phi_p$ is the polymer fraction in the matrix ($\phi_p + \phi_w = 1$), $\chi$ is the Flory-Huggins parameter, and $N$ is the number of monomer units between crosslink junctions. Using parameters reported for PA layer ($\chi \sim 2$, $N \sim 10$)[26,27] and the measured variables in this study ($r_p \sim 0.8$ nm, $\delta r_p/\delta\Delta c \sim 0.2$ nm M⁻¹), Eq. 2 predicts an $\phi_w$ of ~24%, which falls within the reported $\phi_w$ range for PA layer (15%–32%)[28–30]. This remarkable agreement between values deduced

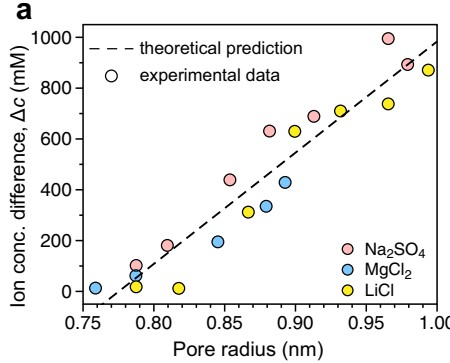

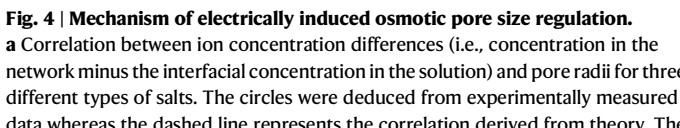

**Fig. 4 | Mechanism of electrically induced osmotic pore size regulation.**
**a** Correlation between ion concentration differences (i.e., concentration in the network minus the interfacial concentration in the solution) and pore radii for three different types of salts. The circles were deduced from experimentally measured data whereas the dashed line represents the correlation derived from theory. The distributions of chemical potenital of water, ion concentration, and hydralic pressure for a conventional (weakly charged) PA membrane (**b**) and an electro-regulated (strongly charged) CNTs-PA membrane (**c**). Source data are provided as a Source Data file.

from experimentally measured data (circles in Fig. 4a) and theoretical correlation derived from first principles (dashed line in Fig. 4a) further supports the theory of pore size regulation via electrically induced osmotic swelling.

For electrolytes that are well rejected by the CNTs-PA membrane, such as $Na_2SO_4$, Eq. 1 also suggests a large concentration gradient within the membrane pores. Because $\triangle c$ is proportional to the interfacial ion concentration, $c_i$, and $c_i$ on the permeate side is nearly two orders of magnitude lower than that on the feed side (Supplementary Table 2), the ion concentration near the pore entrance is also roughly two orders of magnitude higher than that near the pore exit. The distribution of ion concentration through the CNTs-PA matrix also creates a gradient in osmotic pressure that must be balanced by a gradient in hydrostatic pressure, $P_h$. Such gradients of ion concentration, osmotic pressure, and hydrostatic pressure are weaker for electrolytes that are poorly rejected (e.g., LiCl, Supplementary Table 2).

The mechanism of electrically induced osmotic swelling also provides new insights into a more general understanding of mass transport through PA membranes. Under the conventional solution-friction framework developed for RO membranes with strong steric exclusion[31], the ion concentration within the PA is considered to be much lower than that in the solution, i.e., $\triangle c < 0$ (Fig. 4b), which causes PA network deswelling. The interfacial discontinuity of concentration profile arises from ion partition, whereas the interfacial discontinuity of pressure arises from the need to maintain a continuous distribution of the chemical potential of water, $\mu_w$, for steady-state permeation[31].

For strongly charged NF membranes with weaker steric exclusion, such as CNTs-PA membrane when used as a negative electrode, $\triangle c$ can become highly positive (Fig. 4c), which causes polymer network swelling and pore enlargement. To maintain continuity of $\mu_w$, a large hydrostatic pressure should develop inside the PA network accordingly to prevent unlimited infusion of water via osmosis into the polymer matrix. The increased pressure within the PA layer is mechanically balanced by the elastic stress of PA network resulting from network expansion. Such distributions of ion concentration and hydrostatic pressure across a PA layer are unique to highly charged NF membranes and have not been proposed previously. In other words, the electrically induced osmotic impact on pore size is general but only becomes prominent with sizable membrane potential.

In summary, we have demonstrated operando regulation of Angstrom-scale pore size of a conductive PA membrane using a mechanism called electrically induced osmotic swelling. By adjusting the electrical potential imposed on a conductive PA membrane, the intrapore ion concentration and osmotic pressure can be controlled to

induce swelling of the PA polymer network, thereby changing the effective pore size and rejection of neutral molecules. From a practical perspective, this phenomenon of electrically induced osmotic swelling provides a new way for operando pore size regulation that is mechanistically different from the existing strategies of stimuli-responsive pore size modulation and may have potential applications in advanced molecular separation and controlled drug release. From a theoretical perspective, electrically induced osmotic swelling expands our current fundamental understanding on the intricate interactions between solute and membrane and unveils a previously overlooked but important aspect of NF theory.

## Methods

### Preparation of electrically conductive CNTs-PA membrane

The CNTs-PA membrane was fabricated by the interfacial polymerization (IP) of PIP and TMC, following the procedures reported in our previous work[11,12]. First, CNTs nanofilm was prepared by vacuum filtration of hydroxyl-functionalized MWCNTs dispersion onto the surface of the PES MF membrane. Then IP was performed on the surface of the CNTs nanofilm. Specifically, the PES MF membrane was soaked in ethanol for 30 min. Then, 10 ml of 0.1 mg/ml hydroxyl-functionalized MWCNTs dispersion was diluted with ethanol to 100 ml and vacuum-filtrated onto the PES MF membrane at 0.02 MPa vacuum pressure. The prepared CNTs nanofilm was dried in the air for >2 h, then immersed in water for further use. As for the IP, the CNTs nanofilm was placed on the surface of a glass plate. Then, 1.25 mg/ml PIP aqueous solution was used to cover the support surface and stood for 30 s. The excess PIP solution was drained by placing the glass plate vertically until no visible aqueous solution on the membrane surface. Then, 1 mg/ml TMC hexane solution was then poured onto the CNTs support surface for the other 30 s. After removing the excess TMC solution, the membrane was immersed in n-hexane for 30 s to remove unreacted TMC and dried in air at room temperature for 30 min. The resulting membrane was stored in water at 4 °C before use. Besides using CNTs nanofilm as support, we also employed silver nanowires nanofilm as support to prepare the electrically conductive TFC-PA membrane (referred to as silver-PA membrane) following the same IP procedures. And 2-methylpiperazine (MPIP) was also used to replace PIP to prepare the CNTs-PA membrane with reverse polarity following the same IP procedures.

### Characterization

Surface morphology was characterized by a Hitachi S4800 cold field emission scanning electron microscopy with an accelerating voltage of 5 kV and current of 10 mA. Samples were sputter-coated with gold

nanoparticles to inhibit the charging effect. Surface chemical compositions were analyzed using a Thermal Fisher Scientific ESCALAB X-Ray photoelectron spectrometer. XPS specimens were prepared by carefully mounting membranes onto a silicon wafer. XPS peak fitting was performed with XPSPEAK41 software. Cross-sectional TEM images were obtained using a FEI Tecnai G2 F20 S-twin 200 kV field-emission transmission electron microscope. TEM specimens were prepared by embedding the membrane into epoxy resin, then ultrathin slice of the membrane was prepared with a Leika EM UC7/FC7 microtome and carefully mounted onto lacey carbon support grids. Streaming potential measurement was performed on an electro-kinetic Analyzer (SurPASS3, Anton Paar, Ashland, VA) with an adjustable gap cell. The streaming potential values were measured from pH 2 to 10 using 1 mM KCl solution as the background electrolyte at ambient temperature. The concentration of all dye solutions was tested by UV-vis absorption spectrum on a Lambda-25 spectrometer (PerkinElmer Inc., USA). All dye solutions were diluted 10 times before the measurement. Sheet resistance was measured with a Keithley 4200 four-point probe configuration. The membranes were dried in oven at 60 °C before test. Three measurements were made at different locations along the membrane surface. The reported sheet resistance of membranes represented the average of three measurements. The cyclic voltammetry (CV) curve was measured with an electrochemical workstation (Chenhua CHI 660D). The CNTs-PA membrane was firstly immersed in dimethylformamide (DMF) to dissolve the PES MF support. After thoroughly washing the membrane by using DMF, freestanding CNTs-PA layer was transferred onto the surface of nickel foams. Then, the nickel foam was used as an electrode. Through the open circuit potential measurement[32] using titanium mesh as counter electrode, we determined the potential scanning range of CV measurement. The scanning rate was 10 mV s$^{-1}$ in CV measurement.

## Rejection of PEG with different molecular weight by the CNTs-PA membrane

To evaluate the electrical regulation of the pore size of the CNTs-PA membrane, we tested the rejection of PEG with molecular weights of 200, 400, 600, and 1000 Da, respectively, by the CNTs-PA membrane. The feed concentration of PEG aqueous solutions was 200 ppm, and the applied pressure was 2 bar for the measurement. For calculating the rejection $R_{PEG}$, the concentration of feed and permeate was analyzed by a total organic carbon (TOC) equipment (OI Analytical Aurora Model 1030), and the $R_{PEG}$ was calculated according to the below equation

$$R_{PEG} = \frac{C_f - C_p}{C_f} \times 100\% \tag{3}$$

where $C_f$ and $C_p$ represent the concentration of feed and permeate, respectively.

## Pore radius fitted with hindered transport model (HTM)

The NF membrane is modeled as a porous structure with parallel cylindrical pores of uniform size in the HTM[19–21]. Water flux can be modeled using the Hagen-Poiseuille equation:

$$J_v = \frac{r_p^2 (\Delta P - \Delta \pi)}{8 \eta L_e} \tag{4}$$

where $J_v$ is the volumetric permeate flux, $r_p$ is the membrane pore radius, $\eta$ is water dynamic viscosity, $\Delta P$ is hydraulic pressure difference across the membrane, $\Delta \pi$ is osmotic pressure difference between feed and permeate, $L_e$ is effective membrane thickness accounting for porosity $\varepsilon$, tortuosity $\tau$ and the physical (or actual) thickness $L$ via the relation $L_e = \tau L / \varepsilon$.

For uncharged solutes, steric exclusion is the only exclusion mechanism. In the HTM, solutes larger than membrane pores cannot

enter membrane pore. Solutes with size smaller than membrane pores have a size-dependent probability of successful entry (defined as steric exclusion factor, $\phi_{S,i}$ in Eq. (5)), which deviates from the log-normal pore size distribution approach where solutes smaller than membrane pores are always assumed to enter the membrane pore (i.e., a pore entry probability of 100%).

$$\phi_{S,i} = \begin{cases} \left(1 - \frac{r_i}{r_p}\right)^2 for\, r_i < r_p \\ 0, for\, r_i \geq r_p \end{cases} \tag{5}$$

The treatment of size-dependent pore entry probability is more appropriate for NF as the diffusive transport dominates the advective transport for solute partitioning into nanometer or angstrom-level membrane pores. For instance, a Na$^+$ ion has a Peclet number of -10$^{-6}$ for a pore with a radius of 0.8 nm when the permeate flux is -10$^{-5}$ $m\,s^{-1}$, indicating molecular diffusion is the dominant transport mechanism for solutes to approach the membrane pore. The log-normal pore size distribution approach is more suitable for membrane processes with larger pore size, like microfiltration, where advection dominates the solute transport and all solutes can be considered to follow the flow streamlines into the pores.

Solving the HTM leads to the following analytical solution for neutral solute rejection:

$$R_i = 1 - \frac{c_{i,p}}{c_{i,f}} = 1 - \frac{K_{i,a}\phi_{S,i}}{1 - (1 - K_{i,a}\phi_{S,i}) \exp\left(-\frac{J_v L_e}{D_i} \frac{K_{i,a}}{K_{i,d}}\right)} \tag{6}$$

where $D_i$ is solute diffusion efficient in the bulk solution, and $K_{i,a}$ and $K_{i,d}$ are advection and diffusion hindrance factors, which are both functions of the ratio between the solute Stokes radius and pore radius, $\lambda_i = r_i / r_p$. The following empirical correlations, Eqs. 7 and 8, have been used to estimate the hindrance factors.

$$K_{i,a} = \frac{1 + 3.867\lambda_i - 1.907\lambda_i^2 - 0.834\lambda_i^3}{1 + 1.867\lambda_i - 0.741\lambda_i^2} \tag{7}$$

$$K_{i,d} = \begin{cases} \begin{aligned} &1 + \left(\frac{9}{8}\right)\lambda_i ln\lambda_i - 1.56034\lambda_i + 0.528155\lambda_i^2 + 1.951521\lambda_i^3 \\ &\frac{-2.81903\lambda_i^4 + 0.270788\lambda_i^5 + 1.10115\lambda_i^6 - 0.435933\lambda_i^7}{(1-\lambda_i)^2}, \& \lambda_i \leq 0.95 \end{aligned} \\ 0.984\left(\frac{1-\lambda_i}{\lambda_i}\right)^{\frac{5}{2}}, \& \lambda_i > 0.95 \end{cases} \tag{8}$$

Pore radius is then fitted from PEG rejections (MW = 200, 400, 600, 1000 Da) based on Eq. 6.

## Intrapore ion concentration estimated with modified Donnan (mD) model

The mD model assumes a uniform potential distribution throughout the diffuse layer within the membrane pores, i.e., there is a single electrical potential instead of a potential distribution within the pores (but outside the Stern plane). Two potential drops are present in the mD model: the Stern potential, $\triangle\psi_S$, which is the potential drop across the Stern layer; and the Donnan potential, $\triangle\psi_D$, which is a step potential drop across the feed-membrane interface[24,25]. At equilibrium, the concentration of ion species $i$ in the membrane pores ($c_{i,m}$) is related to the Donnan potential ($\triangle\psi_D$) and the feed ion concentration ($c_{i,f}$) according to

$$c_{i,m} = c_{i,f} \exp\left(-\frac{z_i|e|\Delta\psi_D}{kT}\right) \tag{9}$$

Based on the Donnan equilibrium, the partition coefficient of a counter-ion to a matrix, $c_{i,m}/c_{i,f}$, depends on the ion charge valence and the Donnan potential across the interface (Supplementary Fig. 24).

The partition coefficient at a relatively high Donnan potential (e.g., > X mV) can far exceed unity.

The volumetric charge density inside membrane pores, $\sigma_m$, is expressed as

$$\sigma_m = \sum_i z_i c_{i,m} F \tag{10}$$

where $F$ is the Faraday constant. The relationship between $\sigma_m$ and the Stern layer potential and Donnan potential is given by

$$\sigma_m = -C_S \Delta\psi_S = -C_D \Delta\psi_D \tag{11}$$

where $C_S$ and $C_D$ are the volumetric Stern layer capacitance and Donnan layer capacitance, respectively. Overall double-layer capacitance ($C_{DL}$) is expressed as

$$\frac{1}{C_{DL}} = \frac{1}{C_S} + \frac{1}{C_D} \tag{12}$$

At equilibrium, the sum of $\triangle\psi_S$ and $\triangle\psi_D$ equals the actual membrane electrode potential:

$$\psi_m = \Delta\psi_S + \Delta\psi_D \tag{13}$$

The actual membrane electrode potential is measured with a reference Ag/AgCl electrode (Fig. 3b in the main text). The areal overall capacitance is determined from cyclic voltammetry measurements (Supplementary Fig. 15) and converted to volumetric capacitance by normalizing the overall CNTs-PA layer thickness of 500 nm. Intrapore ion concentration can be then estimated by solving Eqs.9–13 to obtain below equation:

$$c_{intra} = \sum_i c_{i,m} \tag{14}$$

### Derivation of the relation between pore size change and increment of intrapore ion concentration

According to the Flory–Rehner theory for a polymer network at swelling equilibrium, osmotic pressure of the external bulk solution equals to the total osmotic pressure of the polymer network[22,23]. The total osmotic pressure of the swollen network has contributions from free energy of mixing, Donnan equilibrium, and polymer elasticity as

$$\pi_b = \pi_{mix} + \pi_{Donnan} + \pi_{elastic} \tag{15}$$

The osmotic pressure due to the free energy of mixing can be expressed based on the Flory–Huggins theory as

$$\frac{\pi_{mix} V_w}{RT} = -In\phi_w - \phi_p - \chi\phi_p^2 \tag{16}$$

where $\chi$ is the Flory–Huggins parameter. The osmotic pressure arising from elasticity for isotropic swelling can be expressed as

$$\frac{\pi_{elastic} V_w}{RT} = -\frac{1}{N}\left(\phi_p^{\frac{1}{3}} - \frac{\phi_p}{2}\right) \tag{17}$$

where $N$ is the number of monomer units between crosslink junctions. By combining Eqs. 15–17, we obtain:

$$\pi_{Donnan} - \pi_b = \frac{RT}{V_w}\left(\ln\phi_w + \phi_p + \chi\phi_p^2 + \frac{1}{N}\left(\phi_p^{\frac{1}{3}} - \frac{\phi_p}{2}\right)\right) \tag{18}$$

Thus, the increment of intrapore osmotic pressure due to the increase of intrapore ion concentration under the applied voltage, $\delta\pi$, can be related to the change of water volume fraction, with the relation $\delta\phi_w = -\delta\phi_p$, by taking the derivative of Eq. 18, we obtain:

$$\delta\pi = -\frac{RT}{V_w}\left(-\frac{1}{\phi_w} + 1 + 2\chi\phi_p + \frac{1}{N}\left(\frac{\phi_p^{-\frac{2}{3}}}{3} - \frac{1}{2}\right)\right)\delta\phi_w \tag{19}$$

Then, assume the water volume fraction is proportional to the cube of the free volume radius, $r$:

$$\phi_w = \alpha r^3 \tag{20}$$

where $\alpha$ is a coefficient related to volumetric pore density and free volume geometry. Thus,

$$\delta\phi_w = 3\alpha r^2 \delta r = 3\frac{\phi_w}{r}\delta r \tag{21}$$

By combining Eqs. 19 and 21, the relation between the increments of effective free volume radius and intrapore ion concentration can be expressed as:

$$\frac{\delta r}{\delta c} = \frac{V_w r}{\phi_w}\left(-\frac{1}{\phi_w} + 1 + 2\chi\phi_p + \frac{1}{N}\left(\frac{\phi_p^{-\frac{2}{3}}}{3} - \frac{1}{2}\right)\right)^{-1} \tag{22}$$

Assume the effective free volume radius equals to the effective pore radius in the hindered transport model, Eq. 22 is the same as Eq. 2.

### Reporting summary
Further information on research design is available in the Nature Portfolio Reporting Summary linked to this article.

## Data availability
All data supporting the findings of this study are available in the article, the Supplementary Information and the Source Data file. Source data are provided with this paper.

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

## Acknowledgements
This work was supported by the National Natural Science Foundation of China (21988102), the National Key Research and Development Plan (2019YFA0705800), the Water Research Foundation (Paul L. Busch Award to S.L.), the US National Science Foundation (2017998), the Key Research and Development Plan of Jiangsu Province (BE2022056), and the Youth Innovation Promotion Association, CAS (2021318).

## Author contributions
Y.Z. and J.J. designed the experiments; S.L. and R.W. developed the theory; L.G. and Y. W. performed the experiments including the fabrication of the electrically conductive PA membranes, characterization, and performance test. R. W. performed the theoretical analysis. All coauthors discussed the results. Y. Z., S. L., J.J., M.E., R.W., and L. G. contributed to writing the manuscript.

## Competing interests
The authors declare no competing interests.
