## [Peer Review File · Nature Communications]

Regulation of Molecular Transport in Polymer Membranes with Voltage-controlled Pore Size at the Angstrom ScaleREVIEWER COMMENTS

Reviewer #1 (Remarks to the Author):

This work presents a method to control the pore radius from 0.8nm to 0.9nm by applying 1V voltage on the Polymer Membranes, and the osmotic pressure is thought to be the reason for the enlargement of the pore. The results are interesting and well organized, it can be published after a major revision. some comments:

- 1) Page 5: it is obviously that the authors make a mistake on the figure number, such as Fig.1e etc should be Fig.2e.
- 2) How many samples have been used in the experiment? How much is the error bar in pore enlargement?
- 3) Table S2 shows that the flux decreases at 1V when MgCl₂ was used, what the possible reason for this?
- 4) The electro-osmotic pressure can enlarge the pore in PA layer, what's the effect of the extra-pressure (1Bar) on the pore size? In general, the applied pressure in the rejection measurement should compress the PA layer, and I guess the electron-osmotic pressure is much less than 2Bar, so it seems the pore radius should be diminished when extra-pressure is applied?
- 5) Currently the pore radius changes from 0.8nm to 0.9nm, is it possible to broaden this range by optimizing the experimental parameters? Like sample material, electrolyte etc.

Reviewer #2 (Remarks to the Author):

Comments for Manuscript NCOMMS-22-48708

Title: Operando Regulation of Molecular Transport in Polymer Membranes with Voltage-controlled Pore Size at the Angstrom Scale

Authors: Yuzhang Zhu et al.

In this manuscript, the authors reported the regulation of the pore size of an electrically conductive polyamide membrane by an applied voltage in the presence of electrolyte via a mechanism that they called the electrically induced osmotic swelling. The results show that, under a proper voltage, the highly charged polyamide layer concentrates counter ions in the polymer network via Donnan equilibrium and creates an osmotic pressure to enlarge the free volume and the effective pore size, and the relation between membrane potential and pore size can be quantitatively described using the extended Flory-Rehner theory with Donnan equilibrium. The strategy and the results presented in this study are interesting and valuable to the membrane community. Therefore, I would like to recommend its publication after careful revisions by addressing the following comments.

1. The authors state that the regulation of the pore size of their electrically conductive polyamide membrane by applied voltage in the presence of electrolyte is based on the mechanism of electrically induced osmotic swelling. To support this statement, although they have presented several indirect evidence including the voltage-dependent rejection of PEG molecules as well as the calculated pore sizes, direct evidence is still necessary. For example, authors should give the experimental data of the voltage-induced swelling of the thickness or dimensions of CNTs-PA membranes in solutions containing 1,000 ppm Na₂SO₄. If the dimensions of CNTs-PA membranes do not change, the voltage-induced swelling of PA in confined spaces in the CNTs-PA membranes should result in a decrease of membrane pore size.
2. To show the potential applications of the proposed membranes as the authors mentioned in the manuscript, some useful applications should be demonstrated, e.g., salt-concentration-responsive permeation adjustment, voltage-dependent molecular separations, and so on.
3. The English language should be polished further.
4. Fig. 3b-3f are not cited properly in the text, and all of them are incorrectly cited as "Fig. 2b-2f".

5. Several colors of the legends in Fig. S8 are very similar and are very difficult to be distinguished.

6. In Fig. S10b, PA cannot be found clearly in the SEM image.

7. In the caption of Fig. 14, the authors state that "The scanning potential range was determined by the open circuit potential measurement shown in Fig.3b in the main text." However, Fig. 3b in the main text shows different data.

Reviewer #3 (Remarks to the Author):

The manuscript by Zhu et al described the fabrication of an electrically conductive CNTs-PA membrane which can be used for solute and molecule separation, with which they found that the pore size in the CNTs-PA layer can be regulated by the applied voltage and the solution concentration. The effect of the electrically induced osmotic swelling (pore size regulation) fitted well with the quantitative model using the extended Flory-Rehner theory. These findings provides very interesting new ideas in the nanofiltration materials and theory, and can have great applications in design and improve the industry technology in water desalination and molecule separation. The manuscript is well written, and the scientific content fits the scope of Nature Communications. The paper is acceptable after some minor corrections.

- 1: the term "operando" seems not necessary shown in the title for the in-situ system used here.
- 2: page 5, line 123, "Fig.2b" should be Fig. 3b.

RESPONSE TO REVIEWERS' COMMENTS

Response to Reviewer #1 (Remarks to the Author)

This work presents a method to control the pore radius from 0.8nm to 0.9nm by applying 1V voltage on the Polymer Membranes, and the osmotic pressure is thought to be the reason for the enlargement of the pore. The results are interesting and well organized, it can be published after a major revision.

Response to general comment: We appreciate the positive comment from the reviewer.

Comment 1: Page 5: it is obviously that the authors make a mistake on the figure number, such as Fig.1e etc should be Fig.2e.

Response: We are sorry for the mistake. We have corrected it in the revised manuscript. We also carefully checked the whole main text to confirm no similar mistakes in the revised manuscript.

Comment 2: How many samples have been used in the experiment? How much is the error bar in pore enlargement?

Response: In this work, every measurement was repeated for more than three times to confirm the reproducibility of research data. And the error bar is no more than 5% for every data. For instance, a largest pore enlargement of 2 Å could be achieved at 1V in the presence of 2000 ppm LiCl as presented in Table S2. The corresponding error bar is around 0.1 Å.

Comment 3: Table S2 shows that the flux decreases at 1V when MgCl₂ was used, what the possible reason for this?

Response: A possible explanation for the flux decreases is the pore blocking induced by the adsorption of Mg²⁺ in the PA chain network. In detail, the high hydration energy of Mg²⁺ (437 kcal mol⁻¹) makes them mainly present in the form of hydrated state in the PA chain network. The large steric hinderance of hydrated Mg²⁺ induces obvious pore block and causes in turn the flux decrease at 1 V. In comparison, the hydration energy of Na⁺ is only 87.2 kcal mol⁻¹, and it is easily dehydrated under pressure. As a result, there is no obvious flux decrease when Na₂SO₄ and NaCl was used. We have added corresponding explanation for the flux decrease below Table S2 in revised

manuscript.

Comment 4: The electro-osmotic pressure can enlarge the pore in PA layer, what's the effect of the extra-pressure (1Bar) on the pore size? In general, the applied pressure in the rejection measurement should compress the PA layer, and I guess the electro-osmotic pressure is much less than 2Bar, so it seems the pore radius should be diminished when extra-pressure is applied?

Response: In this work, the pore enlargement of CNTs-PA membrane was induced by the osmotic swelling, which stems from the potential-induced difference (Δc) in ion concentrations inside and outside the polymer network. Based on the equilibrium, we have estimated the Δc , and the results were shown in Fig. 4a and Table S1. When the CNTs-PA membrane worked as a cathode at 1 V in the presence of 1000 ppm electrolyte solution (e.g., 1000 ppm MgCl_2), the Δc is up to more than 400 mM, which can produce a $\Delta\pi$ around 9.9 bar. Therefore, 1 bar extra-pressure has almost no effect on the pore regulation by the electrically induced osmotic swelling.

Comment 5: Currently the pore radius changes from 0.8nm to 0.9nm, is it possible to broaden this range by optimizing the experimental parameters? Like sample material, electrolyte etc.

Response: In this work, we have studied the effect of applied voltage and the concentration of electrolyte on the pore radius of the CNTs-PA membrane. When voltage increased from 0 V to 1 V, the pore radius increased from around 8.0 Å to 9.1 Å as shown in Fig. 3c. Further increasing the applied voltage (for example 1.5 V) will cause the damage of the PA layer as shown in Fig. S5 and S8. When the concentration of electrolyte (e.g., Na_2SO_4) increased from 500 ppm to 2000 ppm, the pore radius increased from ~8.8 Å to ~9.8 Å as shown in Fig. 3d. Further increasing the salt concentration to a higher value will not increase the pore radius. Therefore, we conclude that it is difficult to broaden the pore radius to a wider range by optimizing the experimental conditions including applied voltage and electrolyte at present. However, we believe that it is possible to broaden the pore range of PA membrane by designing the molecular structure of the monomers with special rigid and large spatial volume. For example, prof. Livingstone and Cooper et al. reported previously the fabrication of a crystalline porous organic cage membrane by using a monomer with special rigid and

spatial volume structure. Their results showed that the specially designed membrane can switch molecular weight cut-off (MWCO) between 600 to 1400 Da under the interaction of solvent-induced phase transition (*Nat. Mater.* **2022**, *21*, 463-470). This is because that when the polymer chain with this special structure rotates, it will display greater spatial size changes.

Response to Reviewer #2 (Remarks to the Author):

In this manuscript, the authors reported the regulation of the pore size of an electrically conductive polyamide membrane by an applied voltage in the presence of electrolyte via a mechanism that they called the electrically induced osmotic swelling. The results show that, under a proper voltage, the highly charged polyamide layer concentrates counter ions in the polymer network via Donnan equilibrium and creates an osmotic pressure to enlarge the free volume and the effective pore size, and the relation between membrane potential and pore size can be quantitatively described using the extended Flory-Rehner theory with Donnan equilibrium. The strategy and the results presented in this study are interesting and valuable to the membrane community.

Response to general comment: We appreciate the positive comment from the reviewer.

Comment 1: The authors state that the regulation of the pore size of their electrically conductive polyamide membrane by applied voltage in the presence of electrolyte is based on the mechanism of electrically induced osmotic swelling. To support this statement, although they have presented several indirect evidence including the voltage-dependent rejection of PEG molecules as well as the calculated pore sizes, direct evidence is still necessary. For example, authors should give the experimental data of the voltage-induced swelling of the thickness or dimensions of CNTs-PA membranes in solutions containing 1,000 ppm Na₂SO₄. If the dimensions of CNTs-PA membranes do not change, the voltage-induced swelling of PA in confined spaces in the CNTs-PA membranes should result in a decrease of membrane pore size

Response: Thanks for the reviewer's suggestion. However, the PA layer is formed near the surface of the ~500 nm-thick CNTs network and imbedded in the CNTs networks. From the cross-sectional TEM images shown in Fig. 2c, the thickness of the PA layer is only tens of nanometers. Therefore, it is almost impossible to in-situ monitor the thickness or dimension change of the CNTs-PA layer. In addition, we propose that the characterizations including the rejection of neutral molecules (e.g., PEG), the ions-

adsorption capacitance of CNTs-PA membrane determined by CV measurement, and the theory calculation are significant to support the mechanism of the pore size regulation via electrically induced osmotic swelling. Our assertion is the following points:

First, Through the rejections of neutral molecules (e.g., PEG) with different molecular weights to determine the pore size of NF membrane is a widely adopted standard method worldwide, which is an in-situ characterization. The in-situ detected enlargement of the MWCO over the increase of the voltage in the presence of electrolyte solution can be counted as strong direct evidence that the applied voltage causes a change in PA membrane pore size. Second, using the CNTs-PA membrane as a working electrode in the presence of electrolyte solution, the applied voltage caused the ion-adsorption in the PA chains is a common phenomenon, which is similar as the charging process of a capacitor. The cyclic voltammetry (CV) measurement revealed the capacitance of the CNTs-PA apparently increase as increasing the concentration of Na₂SO₄ solution at 1 V (Fig. S14). The change trend is consistent with the change of pore radius shown in Fig. 3d. Moreover, the calculated ion concentration difference between inside and outside the CNTs-PA network based on the Donnan equilibrium is linearly correlated to the pore radius.

According to these data presented in the manuscript, we conclude that the regulation of the pore size of CNTs-PA membrane is based on the mechanism of electrically induced osmotic swelling.

Comment 2: To show the potential applications of the proposed membranes as the authors mentioned in the manuscript, some useful applications should be demonstrated, e.g., salt-concentration-responsive permeation adjustment, voltage-dependent molecular separations, and so on.

Response: Indeed, the ability to regulate pore size *via* applied voltage enables the on-off adjustment of salt rejection and permeation during the desalination process. As shown in Fig. R1, the Na₂SO₄ rejection presents a decrease from 98.5% to 90% as switching the voltage from 0 V to 1 V using 1000 ppm Na₂SO₄ as feed. The permeating flux of the CNTs-PA membranes increase from 42 to 65 Lm⁻²h⁻¹ at the same time. The substantial enhancement on the permeating flux is mainly attributed to the enlargement of the pore size. The slight change on the Na₂SO₄ rejection is attributed to the strengthened surface negative potential by the voltage, which partially offsets the

adverse effect of pore enlargement. The change on the rejection and permeating flux can be easily on-off by the voltage as well, as confirmed by the cycle desalination experiment switching between 0 V and 1 V for 10 times (see Fig. R1b). According to the reviewer’s suggestion, we have added the corresponding description “In addition, the ability to regulate pore size via applied voltage enables the on-off adjustment of salt rejection and permeation for desalination. As shown in Supplementary Fig. 14a, the Na_2SO_4 rejection decreased from 98.5% to 90% when the applied voltage changed from 0 V to 1 V using 1000 ppm Na_2SO_4 as feed. The corresponding permeating flux increased from 42 to 65 $\text{Lm}^{-2}\text{h}^{-1}$ simultaneously. The substantial enhancement in the permeating flux is mainly attributed to the enlargement of the pore size. In contrast, the slight change in the Na_2SO_4 rejection is attributed to the enhancement of negative surface potential by the voltage, which partially offsets the adverse effect of pore enlargement. The rejection and permeating flux can be easily on-off tuned by the voltage, as evidenced by switching the voltage between 0 V and 1 V 10 times in the cycling experiment (Supplementary Fig. 14b)” in the revised manuscript (page 7, line 18-28). And Fig. R1 has been added to the revised supporting information as Supplementary Fig. 14.

Fig. R1. (a) The variation of Na_2SO_4 rejection and permeating flux with the increase of the applied voltage when using CNTs-PA membrane as a cathode. (b) The continuous variation of Na_2SO_4 rejection and permeating flux by switching the voltage from 0 V to 1 V in the cycling experiment.

Comment 3: The English language should be polished further.

Response: Thanks for the reviewer’s suggestion. We have polished the language in the revised manuscript.

Comment 4: Fig. 3b-3f are not cited properly in the text, and all of them are incorrectly cited as “Fig. 2b-2f”.

Response: We are sorry for these mistakes and we have corrected them in the revised manuscript.

Comment 5. Several colors of the legends in Fig. S8 are very similar and are very difficult to be distinguished.

Response: We have changed the colors of the legends in Fig. S8 in the revised supporting information to make them easily distinguished.

Supplementary Fig. 8. The rejection of PEG with different molecular weight by the CNTs-PA membrane at different applied voltages in the presence of (a) MgCl₂ and (b) LiCl with different concentrations.

Comment 6: In Fig. S10b, PA cannot be found clearly in the SEM image.

Response: This may be due to the ultrathin thickness of the PA layer formed on the surface silver nanowires nanofilm, the lattice of the silver nanowires nanofilm is clearly observed in the SEM images shown in the Supplementary Fig. 10b. However, if we compare the SEM image of silver-PA membrane with the SEM image of pristine silver nanowires nanofilm, we can also find the apparent difference between them. Because an ultrathin transparent PA layer is covered on the surface of the support, the pore of the support become vague. Meanwhile, the formation of silver-PA layer can be clearly observed from the cross-sectional SEM image shown in Supplementary Fig. 11.

Comment 7: In the caption of Fig. 14, the authors state that “The scanning potential range was determined by the open circuit potential measurement shown in Fig.3b in the

main text.” However, Fig. 3b in the main text shows different data.

Response: We have corrected the mistake and added the data of open circuit potential measurement as Supplementary Fig. 15a in the revised supporting information. We are sorry for this mistake.

Supplementary Fig. 15. (a) Surface potentials of the CNTs-PA membrane vs. the reference electrode (Ag/AgCl) being used it as cathode at different applied voltages in the presence of 1000 ppm Na₂SO₄ aqueous solution. (b) CV curves of the CNTs-PA membrane being used as the cathode in the presence of Na₂SO₄ aqueous solution with different concentrations. A represents the integral area of the CV curve. The scanning potential range was determined by the open circuit potential measurement shown in Supplementary Fig. 15a.

Response to Reviewer #3 (Remarks to the Author):

The manuscript by Zhu et al described the fabrication of an electrically conductive CNTs-PA membrane which can be used for solute and molecule separation, with which they found that the pore size in the CNTs-PA layer can be regulated by the applied voltage and the solution concentration. The effect of the electrically induced osmotic swelling (pore size regulation) fitted well with the quantitative model using the extended Flory-Rehner theory.

These findings provides very interesting new ideas in the nanofiltration materials and theory, and can have great applications in design and improve the industry technology in water desalination and molecule separation. The manuscript is well written, and the scientific content fits the scope of Nature Communications. The paper is acceptable after some minor corrections.

Response: We appreciate the positive comment from the reviewer.

Comment 1: the term “operando” seems not necessary shown in the title for the in-situ system used here.

Response: According to the reviewer’s comment, we delete the word “operando” from the title.

Comment 2: page 5, line 123, “Fig.2b“ should be Fig. 3b.

Response: We are sorry for the mistake. We have corrected it in the revised manuscript. We also carefully checked the whole main text to confirm no similar mistakes in the revised manuscript.

REVIEWER COMMENTS

Reviewer #1 (Remarks to the Author):

the authors have answered my questions, and I recommend the publication of this manuscript in the current version.

Reviewer #2 (Remarks to the Author):

The authors have addressed most of my comments. However, the authors' rebuttal to my comment #1 (about the direct evidence for the pore size change) still hasn't convinced me yet.

In the manuscript and in the rebuttal to my comment #1, all the evidences for the pore size regulation, including the voltage-dependent rejection of PEG molecules as well as the calculated pore sizes, are indirect ones. However, more solid evidence is still needed. The readers DO want to know whether the CNTs networks are flexible or rigid? As I have mentioned, if the CNTs networks are rigid or the thickness or dimension of CNTs-PA layer does not change, the voltage-induced swelling of PA in confined spaces in the CNTs networks should result in a decrease of pore size rather than an increase of pore size.

It is unnecessary to in-situ monitor the thickness or dimension change of the CNTs-PA layer. From the cross-sectional TEM image shown in Fig. 2c, the thickness of the CNTs-PA layer is ~500 nm. If the pore radius of the CNTs-PA layer increases from 0.8 nm to 0.9 nm by applying 1 V voltage, the whole thickness of the CNTs-PA layer should increase several tens of nanometers, which could be easily observed by TEM. Therefore, two rapidly solidified samples of the CNTs-PA membranes respectively representing the two static states before and after applying 1 V voltage are enough for the direct comparison of the whole thickness change of CNTs-PA layer. Such a characterization is not so difficult, but it is especially necessary for supporting the proposed idea in the manuscript.

RESPONSE TO REVIEWERS' COMMENTS

Comment from Reviewer 2:

In the manuscript and in the rebuttal to my comment #1, all the evidences for the pore size regulation, including the voltage-dependent rejection of PEG molecules as well as the calculated pore sizes, are indirect ones. However, more solid evidence is still needed. The readers DO want to know whether the CNTs networks are flexible or rigid? As I have mentioned, if the CNTs networks are rigid or the thickness or dimension of CNTs-PA layer does not change, the voltage-induced swelling of PA in confined spaces in the CNTs networks should result in a decrease of pore size rather than an increase of pore size.

It is unnecessary to in-situ monitor the thickness or dimension change of the CNTs-PA layer. From the cross-sectional TEM image shown in Fig. 2c, the thickness of the CNTs-PA layer is ~500 nm. If the pore radius of the CNTs-PA layer increases from 0.8 nm to 0.9 nm by applying 1 V voltage, the whole thickness of the CNTs-PA layer should increase several tens of nanometers, which could be easily observed by TEM. Therefore, two rapidly solidified samples of the CNTs-PA membranes respectively representing the two static states before and after applying 1 V voltage are enough for the direct comparison of the whole thickness change of CNTs-PA layer. Such a characterization is not so difficult, but it is especially necessary for supporting the proposed idea in the manuscript.

Response:

We thank the reviewer for re-reviewing our manuscript. We agree with the reviewer that direct observation of dimensional change of the swollen active layer would provide the definitive proof of the proposed idea. However, performing the experiment proposed by the reviewer is difficult and not capable of providing the intended information for the reasons below:

(1) The PA layer is not 500 nm thick but tens of nm thick as shown in Fig.2a and 2c. ~500 nm is the thickness of the CNT network, but the PA layer only exists at the top in the CNT network. The dimensional change on the PA thickness induced by the swelling of polymer network is approximate several of nm. Additionally, the swelling of PA network is induced by the applied voltage. If we remove the voltage, the swelling will disappear, and the pore size of PA layer will

recover to the initiate state. Without a in-situ characterization technology, it is almost impossible to observe the dimensional change in PA layer by TEM.

(2) Let us now discuss the feasibility of the proposed experiment. If by “rapidly solidify” the reviewer meant freeze dry, there are at least two problems. First, operationally, it is not clear how one can perform freeze dry of an example with a voltage applied to it in a solution. Second, freezing itself will induce swelling of a polymer network with water as ice has a higher molar volume than liquid water (which is why outdoor water hoses could break in freezing weather if undrained). What may (or may not) further complicates the problem is the effect of solute exclusion during freezing.

REVIEWER COMMENTS

Reviewer #2 (Remarks to the Author):

As I have mentioned, the readers DO want to know whether the CNTs networks are FLEXIBLE OR RIGID?

If the CNTs networks are RIGID, the voltage-induced swelling of PA in confined spaces in the CNTs networks should result in a DECREASE of pore size rather than an INCREASE of pore size, which is a key point of this manuscript.

I hope the authors will try to provide some evidence to demonstrate the flexibility of the CNTs networks.

Reviewer #4 (Remarks to the Author):

Nat commun review

Manuscript # NCOMMS-22-48708B

Title Regulation of Molecular Transport in Polymer Membranes with Voltage-controlled Pore Size at the Angstrom Scale

The authors present an in-situ electric field induced separation enhancement in liquid environment of polyamide membranes. The proposed mechanism is the ion movement inside the polymer, that leads to swelling and relaxation of the film upon polarization/depolarization.

Driving ions through a membrane by applying a voltage is not a novel concept, but actually a very easy one and here, the authors present a highly interesting study on a never observed effect. To the best of my knowledge, nobody ever looked at the swelling and deswelling of the "solid electrolyte" – what a membrane could also be seen as.

I see the work here a bit different to what the authors see as the context. They put CNTs to one side of the membrane to act as an electrode, while the PA membrane itself acts as capacitive layer. The counter electrode then builds up charges that attract the ions and drives the separation. Combining this with the swelling and deswelling through electric field influence makes more sense. I highly recommend a rethinking of the mechanistic explanation and a rephrasing and would then definitely like to see this study published in N Comms.

A few questions arise – not on the way the experiments are performed – this is done very nicely - but on the interpretation of the results.

1) If I see this correctly, the polymer itself is non-conductive but highly polarizable. You add a conductive layer of CNTs to it, to get the back/front-side conductive. Comparing your membrane to an electrolyte in batteries makes more sense, than talking about a conductive membrane. The membrane itself is a capacitive layer and will lead to charge accumulation, while the ions are mobile inside the film and get pulled towards the counter electrode. Compare this here: <https://doi.org/10.1016/j.jpowsour.2015.06.028>

2) In Li-ion batteries, swelling of all components is a common thing (and problem). You mention COF membranes, because they are more near to what you think a polymer is. However, the phenomenon polarizable framework is more a thing in metal-organic frameworks (MOFs). Here, it is possible to achieve polymorphic phase changes in crystals using electric fields <https://dx.doi.org/10.1126/science.aal2456> - which is somehow the same effect that you describe here – so to say, a swelling and deswelling in a crystalline material. I see many parallels. Your explanations are very well suited, but to the best of my understanding, conductivity is not a thing in your materials, but polarizability is.

3) While the ion mobility controlled by the electrodes – CNT and counter-electrode – is a very common concept, the swelling and deswelling by external stimuli in water purification membranes has not yet been investigated to the best of my knowledge. This is very interesting. I would re-

think the hypothesis of the "conductivity" in the membrane. The ions are moving, which is some kind of conductivity, but the polymer itself is not conductive. Maybe the authors rethink the proposed concept to electric field induced phenomena, rather than conductivity induced phenomena. Maybe I am wrong – then please explain to me.

4) You have a charged polymer network, which can be distorted in an electric field, as the chains are very mobile. I think the hypothesis of an electrical actuating sponge fits quite nicely. The fact that the effect is reversible speaks for my hypothesis as well.

RESPONSE TO REVIEWERS' COMMENTS

The authors present an in-situ electric field induced separation enhancement in liquid environment of polyamide membranes. The proposed mechanism is the ion movement inside the polymer that leads to swelling and relaxation of the film upon polarization/depolarization. Driving ions through a membrane by applying a voltage is not a novel concept, but actually a very easy one and here, the authors present a highly interesting study on a never observed effect. To the best of my knowledge, nobody ever looked at the swelling and deswelling of the “solid electrolyte” – what a membrane could also be seen as.

I see the work here a bit different to what the authors see as the context. They put CNTs to one side of the membrane to act as an electrode, while the PA membrane itself acts as capacitive layer. The counter electrode then builds up charges that attract the ions and drives the separation. Combining this with the swelling and deswelling through electric field influence makes more sense.

I highly recommend a rethinking of the mechanistic explanation and a rephrasing and would then definitely like to see this study published in N Comms.

Our response: We first like to sincerely thank the reviewer for the overall positive affirmation of the novelty of our work. Based on the reviewer’s overall suggestion and description, we think that the reviewer might have misinterpreted the configuration of our system, especially the position of the PA layer.

The CNTs network itself did not have the function of separation for molecules in the range of molecular weight. The real “membrane” that serves as the active separation layer was the PA layer. In the system reported in this study, the PA layer is embedded near the surface the CNTs network to be used as a working electrode and should thus have the same electrical potential as the CNTs network (see the schematic below).

Also, the reviewer’s mentioning of “Driving ions through a membrane by applying a voltage” or “the counter electrode then builds up charges that attract the ions and drives the separation” and referring to the membrane as “solid electrolyte” suggest to us that the reviewer may misinterpreted the system as similar to an electrochemical energy storage (e.g., battery) system which is not the case. In our nanofiltration system, ions are not moving through the membrane (either the PA active layer, the CNTs network, or the macroporous support layer) under applied electric field and there was no electrode reaction to sustain any current. The transport of molecules through the PA layer was driven primarily by diffusion due to the concentration gradient and by advection due to the pressure-driven flow. The only current that may had occurred was due to capacitive charging of the CNTs network and the PA layer upon the application of the voltage, but the capacity was very small compared to any electrochemical energy storage systems.

We provide illustrative comparison between what we speculate as the reviewer’s interpretation (left) and what the system really was (right) to hopefully make it clearer.

A few questions arise – not on the way the experiments are performed – this is done very nicely - but on the interpretation of the results.

1) If I see this correctly, the polymer itself is non-conductive but highly polarizable. You add a conductive layer of CNTs to it, to get the back/front-side conductive. Comparing your membrane to an electrolyte in batteries makes more sense, than talking about a conductive membrane. The membrane itself is a capacitive layer and will lead to charge accumulation, while the ions are mobile inside the film and get pulled towards the counter electrode. Compare this here: <https://doi.org/10.1016/j.jpowsour.2015.06.028>

Our response: Again, we would like to clarify that the polymer itself (e.g. the PA layer with a thickness at the order of 10 nm) was completely embedded in a 500 nm-thick CNTs network (see Figure 2 in the manuscript, or copied below). The complete PA layer should have the same potential as CNTs network and the electrolyte it contacts. Thus, the PA layer has no similarity to a solid electrolyte in battery.

2) In Li-ion batteries, swelling of all components is a common thing (and problem). You mention COF membranes, because they are more near to what you think a polymer is. However, the phenomenon polarizable framework is more a thing in metal-organic frameworks (MOFs). Here,

it is possible to achieve polymorphic phase changes in crystals using electric fields <https://dx.doi.org/10.1126/science.aal2456> - which is somehow the same effect that you describe here – so to say, a swelling and deswelling in a crystalline material. I see many parallels. Your explanations are very well suited, but to the best of my understanding, conductivity is not a thing in your materials, but polarizability is.

Our response: We thank the reviewer for pointing out an interesting phenomenon that we were unaware of. However, again, based on our clarification above, the active layer responsible for the separation was not sandwiched between the two electrodes but rather completely embedded in the CNTs network as a working electrode and has the same potential as the contacting current collector.

While there was electric field between the two electrodes across the bulk solution, there is no electric field in the PA layer but only a uniform potential. Thus, our system is not analogous to the system in the reference provided by the reviewer.

3) While the ion mobility controlled by the electrodes – CNT and counter-electrode – is a very common concept, the swelling and deswelling by external stimuli in water purification membranes has not yet been investigated to the best of my knowledge. This is very interesting. I would re-think the hypothesis of the “conductivity” in the membrane. The ions are moving, which is some kind of conductivity, but the polymer itself is not conductive. Maybe the authors rethink the proposed concept to electric field induced phenomena, rather than conductivity induced phenomena. Maybe I am wrong – then please explain to me.

Our response: We would like to provide a brief summary of what we think had happened which was also supported by multiple pieces of experimental evidence and also the quantitative theoretical model we developed.

The very thin (10-50 nm) PA layer embedded in the CNTs network, though not very conductive by itself, can now have the same potential of the 500 nm-thick CNTs network. Now the PA layer serves as a capacitor and thus attracts counter ions into its very small pores (just as in super capacitor). However, the specific capacitance is very small (as measured) and the purpose was NOT to store ions. Rather, concentrating counter- ions to its pores will increase the osmotic pressure of the solution within the PA layer pores and thus swell those pores and make PA layer less rejective to (neutral) molecules.

The multiple pieces of evidence that support this theory include:

- If there was no salt, then applying voltage did not lead to swelling (because no osmotic pressure without salt).
- If the membrane capacitance was too small (i.e., little capacitive salt adsorption), then the swelling effect was also very weak.
- The degree of swelling could be well described by the Flory-Rehner-Donnan model.

4) You have a charged polymer network, which can be distorted in an electric field, as the chains are very mobile. I think the hypothesis of an electrical actuating sponge fits quite nicely. The fact that the effect is reversible speaks for my hypothesis as well.

Our response: As we mentioned above and wrote also very clearly in the manuscript (with excerpt copied below), the swelling was most likely an osmotic effect because (1) it did NOT occur with an applied voltage if there was NO salt, and (2) its degree is clearly dependent on electrolyte concentration. Additionally, we have also discussed above that there was no electric field in the PA layer that was responsible for the separation.

“Moreover, when the feed solution contains no electrolyte, applied voltage had almost no impact on effective pore size (Fig. S23), which suggests that the electrically driven swelling is not a Coulombic effect but relies on osmotic pressure and requires the presence of electrolyte. In fact, the strong dependence of the effective pore size on electrolyte concentration (Fig. 3d) also supports the mechanism of electrically induced osmotic swelling.”

Figure S23 (a). The rejection of PEG with different molecular weight at 0 and 1V with pure water (no electrolyte in the solution). The two curves completely overlap, i.e., no difference of rejection was observed.

Figure 3(d). Pore radius of CNTs-PA membrane as a function of Na₂SO₄ concentration at an applied voltage of 1.0 V.

REVIEWERS' COMMENTS

Reviewer #4 (Remarks to the Author):

The authors answered all my questions satisfactorily. Even though we seem not to be on the same level, I think we interpret it the same way. My expertise lies more on the level of inorganic materials, therefore our mechanistic understanding seems to be different. Especially the answer to my comment 3 makes me happy, as we seem to have more or less the same idea of the system. The interpretation of the authors seems very well thought through and their explanations make sense. I am still a bit of a different opinion on the role of the polymeric membrane and CNT network. The CNT surrounding the polymer will still build up a capacitive effect inside the PA layer, just for the sake of the polymer being a dielectric and the CNTs being charged. However, this does not make a difference to the underlying, great science and the obtained results. Science is made for discussion.

I hope we can expect more investigations on the matter after this study is published, as this is really a breakthrough for membrane separation.